# Unsupervised Adversarial Invariance

**Ayush Jaiswal, Yue Wu, Wael AbdAlmageed, Premkumar Natarajan**
USC Information Sciences Institute
Marina del Rey, CA, USA
{ajaiswal, yue_wu, wamageed, pnataraj}@isi.edu

## Abstract

Data representations that contain all the information about target variables but are invariant to nuisance factors benefit supervised learning algorithms by preventing them from learning associations between these factors and the targets, thus reducing overfitting. We present a novel unsupervised invariance induction framework for neural networks that learns a split representation of data through competitive training between the prediction task and a reconstruction task coupled with disentanglement, without needing any labeled information about nuisance factors or domain knowledge. We describe an adversarial instantiation of this framework and provide analysis of its working. Our unsupervised model outperforms state-of-the-art methods, which are supervised, at inducing invariance to inherent nuisance factors, effectively using synthetic data augmentation to learn invariance, and domain adaptation. Our method can be applied to any prediction task, eg., binary/multi-class classification or regression, without loss of generality.

## 1 Introduction

Supervised learning, arguably the most popular branch of machine learning, involves estimating a mapping from data samples ($x$) to target variables ($y$). A common formulation of this task is the estimation of the conditional probability $p(y|x)$ from data through learning associations between $y$ and underlying factors of variation of $x$. However, data often contains nuisance factors ($z$) that are irrelevant to the prediction of $y$ from $x$ and estimation of $p(y|x)$ in such cases leads to overfitting when the model *incorrectly* learns to associate some $z$ with $y$. Thus, when applied to new data containing unseen variations of $z$, trained models perform poorly. For example, a nuisance factor in the case of face recognition in images is the lighting condition the photograph was captured in, and a recognition model that associates lighting with subject identity is expected to perform poorly. Developing machine learning methods that are *invariant* to nuisance factors has been a long-standing problem in machine learning; studied under various names such as "feature selection", "robustness through data augmentation" and "invariance induction".

While deep neural networks (DNNs) have outperformed traditional methods at highly sophisticated and challenging supervised learning tasks, providing better estimates of $p(y|x)$, they are prone to the same problem of incorrectly learning associations between $z$ and $y$. An architectural solution to this problem is the development of neural network units that capture specific forms of information, and thus are inherently invariant to certain nuisance factors [3, 19]. For example, convolutional operations coupled with pooling strategies capture shift-invariant spatial information while recurrent operations robustly capture high-level trends in sequential data. However, this approach requires significant effort for *engineering* custom network modules and layers to achieve invariance to *specific* nuisance factors, making it inflexible [19]. A different but popularly adopted solution to the problem of nuisance factors is the use of data augmentation where synthetic versions of real data samples are generated, during training, with *specific* forms of variation [3]. For example, rotation, translation and additive noise are typical methods of augmentation used in computer vision, especially for

classification and detection tasks. However, models trained naïvely on the augmented dataset become robust to *limited* forms of nuisance by learning to associate every *seen* variation of such factors to the target variables. Consequently, such models perform poorly when applied to data exhibiting *unseen* nuisance variations, such as face images at previously unseen pose angles.

A related but more systematic solution to this problem is the approach of invariance-induction by guiding neural networks through specialized training mechanisms to discard *known* nuisance factors from the learned latent representation of data that is used for prediction. Models trained in this fashion become robust by exclusion rather than inclusion and are, therefore, expected to perform well even on data containing variations of specific nuisance factors that were not seen during training. For example, a face recognition model trained explicitly to not associate lighting conditions with the identity of the person is expected to be more robust to lighting conditions than a similar model trained naïvely on images of subjects under *certain* different lighting conditions [19]. This research area has, therefore, garnered tremendous interest recently [6, 13, 14, 19]. However, a shortcoming of this approach, is the requirement of domain knowledge of possible nuisance factors and their variations, which is often hard to find [3]. Additionally, this solution to invariance applies only to cases where annotated data is available for each nuisance factor, such as labeled information about the lighting condition of each image in the face recognition example, which is often not the case.

We present a novel unsupervised framework for invariance induction that overcomes the drawbacks of previous methods. Our framework promotes invariance through separating the underlying factors of variation of $x$ into two latent embeddings: $e_1$, which contains all the information required for predicting $y$, and $e_2$, which contains other information irrelevant to the prediction task. While $e_1$ is used for predicting $y$, a noisy version of $e_1$, denoted as $\tilde{e}_1$, and $e_2$ are used to reconstruct $x$. This creates a *competitive* scenario where the reconstruction module tries to pull information into $e_2$ (because $\tilde{e}_1$ is unreliable) while the prediction module tries to pull information into $e_1$. The training objective is augmented with a disentanglement term that ensures that $e_1$ and $e_2$ do not contain redundant information. In our adversarial instantiation of this generalized framework, disentanglement is achieved between $e_1$ and $e_2$ in a novel way through two adversarial *disentanglers* — one that aims to predict $e_2$ from $e_1$ and another that does the inverse. The parameters of the combined model are learned through adversarial training between (a) the encoder, the predictor and the decoder, and (b) the disentanglers. The framework makes no assumptions about the data, so it can be applied to any prediction task without loss of generality, be it binary/multi-class classification or regression. Unlike existing methods, the proposed method does not require annotation of nuisance factors or specialized domain knowledge. We provide results on three tasks involving a diverse collection of datasets — (1) invariance to inherent nuisance factors, (2) effective use of synthetic data augmentation for learning invariance and (3) domain adaptation. Our unsupervised framework outperforms existing approaches for invariance induction, which are supervised, on all of them.

## 2   Related Work

Methods for preventing supervised learning algorithms from learning false associations between target variables and nuisance factors have been studied from various perspectives including "feature selection" [16], "robustness through data augmentation" [10, 11] and "invariance induction" [3, 14, 19]. Feature selection has typically been employed when data is available as a set of conceptual features, some of which are irrelevant to the prediction tasks. Our approach can be interpreted as an implicit feature selection mechanism for neural networks, which can work on both raw data (such as images) and feature-sets (e.g., frequency features computed from raw text). Popular feature selection methods [16] incorporate information-theoretic measures or use supervised methods to score features with their importance for the prediction task and prune the low-scoring ones. Our framework performs this task implicitly on latent features that the model learns by itself from the provided data.

Deep neural networks (DNNs) have outperformed traditional methods at several supervised learning tasks. However, they have a large number of parameters that need to be estimated from data, which makes them especially vulnerable to learning relationships between target variables and nuisance factors and, thus, overfitting. The most popular approach to expand the data size and prevent overfitting in deep learning has been synthetic data augmentation [3, 5, 9–11], where multiple copies of data samples are created by altering variations of certain known nuisance factors. DNNs trained with data augmentation have been shown to generalize better and be more robust compared to those trained without in many domains including vision, speech and natural language. This approach

works on the principle of inclusion. More specifically, the model learns to associate multiple seen variations of those nuisance factors to each target value. In contrast, our method encourages exclusion of information about nuisance factors from latent features used for predicting the target, thus creating more robust features. Furthermore, combining our method with data augmentation further helps our framework remove information about nuisance factors used to synthesize additional data, without the need to explicitly quantify or annotate the generated variations. This is especially helpful in cases where augmentation is performed using sophisticated analytical or composite techniques.

Several supervised methods for invariance induction and invariant feature learning have been developed recently, such as Controllable Adversarial Invariance (CAI) [19], Variational Fair Autoencoder (VFAE) [14], and a maximum mean discrepancy based model (NN+MMD) [13]. These methods use annotated information about variations of specific nuisance factors to force their exclusion from the learned latent representation. They have also been applied to learn "fair" representations based on domain knowledge, such as making predictions about the savings of a person invariant to age, where making the prediction task invariant to such factors is of higher priority than the prediction performance itself [19]. Our method induces invariance to nuisance factors with respect to a supervised task in an unsupervised way. However, it is not guaranteed to work in "fairness" settings because it does not incorporate any external knowledge about factors to induce invariance to.

Disentangled representation learning is closely related to our work since disentanglement is one of the pillars of invariance induction in our framework as the model learns two embeddings (for any given data sample) that are expected to be uncorrelated to each other. Our method shares some properties with multi-task learning (MTL) [17] in the sense that the model is trained with multiple objectives. However, a fundamental difference between our framework and MTL is that the latter promotes a shared representation across tasks whereas the only information shared *loosely* between the tasks of predicting $y$ and reconstructing $x$ in our framework is a noisy version of $e_1$ to help reconstruct $x$ when combined with a separate encoding $e_2$, where $e_1$ itself is used directly to predict $y$.

## 3    Unsupervised Adversarial Invariance

In this section, we describe a generalized framework for unsupervised induction of invariance to nuisance factors by disentangling information required for predicting $y$ from other unrelated information contained in $x$ through the incorporation of data reconstruction as a competing task for the primary prediction task and a disentanglement term in the training objective. This is achieved by learning a split representation of data as $e = [e_1 \ e_2]$, such that information essential for the prediction task is pulled into $e_1$ while all other information about $x$ migrates to $e_2$. We present an adversarial instantiation of this framework, which we call Unsupervised Adversarial Invariance.

### 3.1    Unsupervised Invariance Induction

Data samples ($x$) can be abstractly represented as a set of underlying factors of variation $F = \{f_i\}$. This can be as simple as a collection of numbers denoting the position of a point in space or as complicated as information pertaining to various facial attributes that combine non-trivially to form the image of someone's face. Understanding and modeling the interactions between factors of variation of data is an open problem. However, supervised learning of the mapping of $x$ to target ($y$) involves a relatively simpler (yet challenging) problem of finding those factors of variation ($F_y$) that contain all the information required for predicting $y$ and discarding all the others ($\overline{F}_y$). Thus, $F_y$ and $\overline{F}_y$ form a partition of $F$, where we are more interested in the former than the latter. Since $y$ is independent of $\overline{F}_y$, i.e., $y \perp \overline{F}_y$, we get $p(y|x) = p(y|F_y)$. Estimating $p(y|x)$ as $q(y|F_y)$ from data is beneficial because the nuisance factors, which comprise $\overline{F}_y$, are never presented to the estimator, thus avoiding inaccurate learning of associations between nuisance factors and $y$. This forms the basis for "feature selection", a research area that has been well-studied.

We incorporate the idea of splitting $F$ into $F_y$ and $\overline{F}_y$ in our framework in a more relaxed sense as learning a disentangled latent representation of $x$ in the form of $e = [e_1 \ e_2]$, where $e_1$ aims to capture all the information in $F_y$ and $e_2$ that in $\overline{F}_y$. Once trained, the model can be used to infer $e_1$ from $x$ followed by $y$ from $e_1$. More formally, our general framework for unsupervised invariance induction comprises four core modules: (1) an encoder $Enc$ that embeds $x$ into $e = [e_1 \ e_2]$, (2) a predictor $Pred$ that infers $y$ from $e_1$, (3) a noisy-transformer $\psi$ that converts $e_1$ into its noisy version $\tilde{e}_1$, and

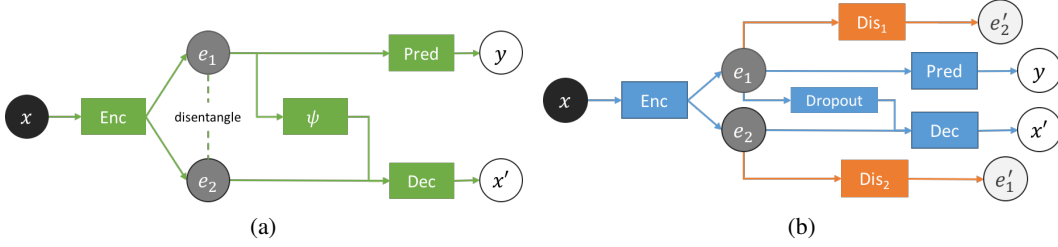

Figure 1: (a) Unsupervised Invariance Induction Framework and (b) Adversarial Model Design

(4) a decoder $Dec$ that reconstructs $x$ from $\tilde{e}_1$ and $e_2$. Additionally, the training objective contains a loss-term that enforces disentanglement between $Enc(x)_1 = e_1$ and $Enc(x)_2 = e_2$. Figure 1a shows our generalized framework. The training objective for this system can be written as Equation 1.

$$
\begin{aligned}
L &= \alpha L_{pred}(y, Pred(e_1)) + \beta L_{dec}(x, Dec(\psi(e_1), e_2)) + \gamma L_{dis}((e_1, e_2)) \\
&= \alpha L_{pred}(y, Pred(Enc(x)_1)) + \beta L_{dec}(x, Dec(\psi(Enc(x)_1), Enc(x)_2)) + \gamma L_{dis}(Enc(x)) \quad (1)
\end{aligned}
$$

The predictor and the decoder are designed to enter into a *competition*, where $Pred$ tries to pull information relevant to $y$ into $e_1$ while $Dec$ tries to extract all the information about $x$ into $e_2$. This is made possible by $\psi$, which makes $\tilde{e}_1$ an unreliable source of information for reconstructing $x$. Moreover, a version of this framework without $\psi$ can converge to a degenerate solution where $e_1$ contains all the information about $x$ and $e_2$ contains nothing (noise), because absence of $\psi$ allows $e_1$ to be readily available to $Dec$. The competitive pulling of information into $e_1$ and $e_2$ induces information separation such that $e_1$ tends to contain more information relevant for predicting $y$ and $e_2$ more information irrelevant to the prediction task. However, this competition is not sufficient to completely partition information of $x$ into $e_1$ and $e_2$. Without the disentanglement term ($L_{dis}$) in the objective, $e_1$ and $e_2$ can contain redundant information such that $e_2$ has information relevant to $y$ and, more importantly, $e_1$ contains nuisance factors. The disentanglement term in the training objective encourages the desired clean partition. Thus, *essential* factors required for predicting $y$ concentrate into $e_1$ and all other factors migrate to $e_2$.

## 3.2 Adversarial Model Design and Optimization

While there are numerous ways to implement the proposed unsupervised invariance induction framework, we adopt an adversarial model design, introducing a novel approach to disentanglement in the process. $Enc$, $Pred$ and $Dec$ are modeled as neural networks. $\psi$ can be modeled as a parametric noisy-channel, where the parameters of $\psi$ can also be learned during training. However, we model $\psi$ as dropout [18] (multiplicative Bernoulli noise) because it provides an easy and straightforward method for noisy-transformation of $e_1$ into $\tilde{e}_1$ without complicating the training process.

We augment these core modules with two adversarial *disentanglers* $Dis_1$ and $Dis_2$. While $Dis_1$ aims to predict $e_2$ from $e_1$, $Dis_2$ aims to do the inverse. Hence, their objectives are in direct opposition to the desired disentanglement, forming the basis for adversarial minimax optimization. Thus, $Enc$, $Pred$ and $Dec$ can be thought of as a composite model ($M_1$) that is pit against another composite model ($M_2$) containing $Dis_1$ and $Dis_2$. Figure 1b shows our complete model design with $M_1$ represented by the color blue and $M_2$ with orange. The model is trained end-to-end through backpropagation by playing the minimax game described in Equation 2.

$$
\begin{aligned}
\min_{Enc, Pred, Dec} \; &\max_{Dis_1, Dis_2} \; J(Enc, Pred, Dec, Dis_1, Dis_2); \;\; \text{where:} \\
J(Enc, Pred, &Dec, Dis_1, Dis_2) \\
&= \alpha L_{pred}\big(y, Pred(e_1)\big) + \beta L_{dec}\big(x, Dec(\psi(e_1), e_2)\big) + \gamma \tilde{L}_{dis}\big((e_1, e_2)\big) \\
&= \alpha L_{pred}\big(y, Pred(Enc(x)_1)\big) + \beta L_{dec}\big(x, Dec(\psi(Enc(x)_1), Enc(x)_2)\big) \\
&\quad + \gamma \big\{ \tilde{L}_{dis_1}\big(Enc(x)_2, Dis_1(Enc(x)_1)\big) + \tilde{L}_{dis_2}\big(Enc(x)_1, Dis_2(Enc(x)_2)\big) \big\} \quad (2)
\end{aligned}
$$

We use mean squared error for the disentanglement losses $\tilde{L}_{dis_1}$ and $\tilde{L}_{dis_2}$. We optimize the proposed adversarial model using a scheduled update scheme where we freeze the weights of a composite

| Metric | NN + MMD [13] | VFAE [14] | CAI [19] | Ours |
|---|---|---|---|---|
| Accuracy of predicting $y$ from $e_1$ ($A_y$) | 0.82 | 0.85 | 0.89 | **0.95** |
| Accuracy of predicting $z$ from $e_1$ ($A_z$) | - | 0.57 | 0.57 | **0.24** |

Table 1: Results on Extended Yale-B dataset

player model ($M_1$ or $M_2$) when we update the weights of the other. $M_2$ should ideally be trained to convergence before updating $M_1$ in each training epoch to backpropagate accurate and stable disentanglement-inducing gradients to $Enc$. However, this is not scalable in practice. We update $M_1$ and $M_2$ in the frequency ratio of $1 : k$. We found $k = 5$ to perform well in our experiments.

## 4 Analysis

**Competition between prediction and reconstruction.** The prediction and reconstruction tasks in our framework are designed to compete with each other. Thus, $\eta = \frac{\alpha}{\beta}$ influences which task has higher priority in the overall objective. We analyze the affect of $\eta$ on the behavior of our framework at optimality, considering perfect disentanglement of $e_1$ and $e_2$. There are two asymptotic scenarios with respect to $\eta$: (1) $\eta \to \infty$ and (2) $\eta \to 0$. In case (1), our framework reduces to a predictor model, where the reconstruction task is completely disregarded. Only the branch $x \dashrightarrow e_1 \dashrightarrow y$ remains functional. Consequently, $e_1$ contains all $f \in F'$ at optimality, where $F_y \subseteq F' \subseteq F$. In contrast, case (2) reduces the framework to an autoencoder, where the prediction task is completely disregarded, and only the branch $x \dashrightarrow e_2 \dashrightarrow x'$ remains functional because the other input to $Dec$, $\psi(e_1)$, is noisy. Thus, $e_2$ contains all $f \in F$ and $e_1$ contains nothing at optimality, under perfect disentanglement. In transition from case (1) to case (2), by keeping $\alpha$ fixed and increasing $\beta$, the reconstruction loss starts contributing more to the overall objective, thus inducing more competition between the two tasks. As $\eta$ is gradually decreased, $f \in (F' \setminus F_y) \subseteq \overline{F}_y$ migrate from $e_1$ to $e_2$ because $f \in \overline{F}_y$ are irrelevant to the prediction task but can improve reconstruction by being more readily available to $Dec$ through $e_2$ instead of $\psi(e_1)$. After a point, further decreasing $\eta$ is, however, detrimental to the prediction task as the reconstruction task starts dominating the overall objective and pulling $f \in F_y$ from $e_1$ to $e_2$.

**Equilibrium analysis of adversarial instantiation.** The disentanglement and prediction objectives in our adversarial model design can simultaneously reach an optimum where $e_1$ contains $F_y$ and $e_2$ contains $\overline{F}_y$. Hence, the minimax objective in our method has a *win-win equilibrium*.

**Selecting loss weights.** Using the above analyses, any $\gamma$ that successfully disentangles $e_1$ and $e_2$ should be sufficient. On the other hand, $\alpha$ and $\beta$ can be selected by starting with $\alpha \gg \beta$ and gradually increasing $\beta$ as long as the performance of the prediction task improves. We found $\alpha = 100$, $\beta = 0.1$ and $\gamma = 1$ to work well for all datasets on which we evaluated the proposed model.

## 5 Experimental Evaluation

We provide experimental results on three tasks relevant to invariant feature learning for improved prediction of target variables: (1) invariance to inherent nuisance factors, (2) effective use of synthetic data augmentation for learning invariance, and (3) domain adaptation through learning invariance to "domain" information. We evaluate the performance of our model and prior works on two metrics – accuracy of predicting $y$ from $e_1$ ($A_y$) and accuracy of predicting $z$ from $e_1$ ($A_z$). The goal of the model is to achieve high $A_y$ and $A_z$ close to random chance.

### 5.1 Invariance to inherent nuisance factors

We provide results of our framework at the task of learning invariance to inherent nuisance factors on two datasets – Extended Yale-B [7] and Chairs [2].

**Extended Yale-B.** This dataset contains face-images of 38 subjects under various lighting conditions. The target $y$ is the subject identity whereas the inherent nuisance factor $z$ is the lighting condition. We compare our framework to existing state-of-the-art supervised invariance induction methods, CAI [19], VFAE [14], and NN+MMD [13]. We use the prior works' version of the dataset, which has lighting conditions classified into five groups – front, upper-left, upper-right, lower-left

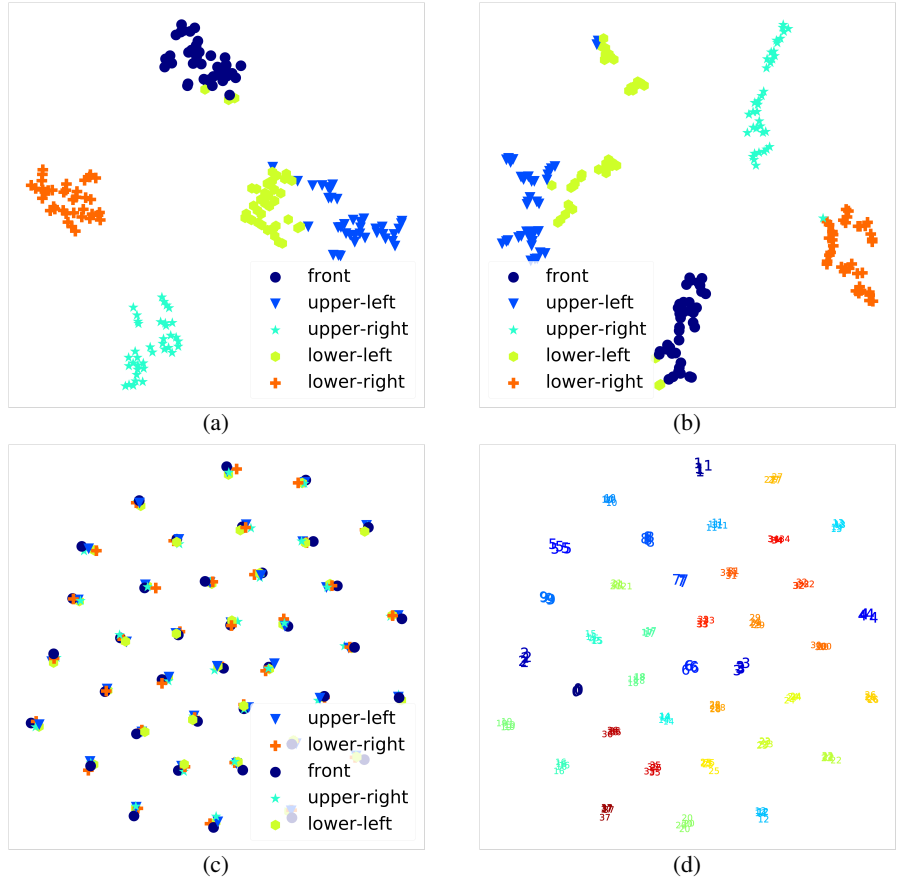

Figure 2: Extended Yale-B – t-SNE visualization of (a) raw data, (b) $e_2$ labeled by lighting condition, (c) $e_1$ labeled by lighting condition, and (d) $e_1$ labeled by subject-ID (numerical markers, not colors).

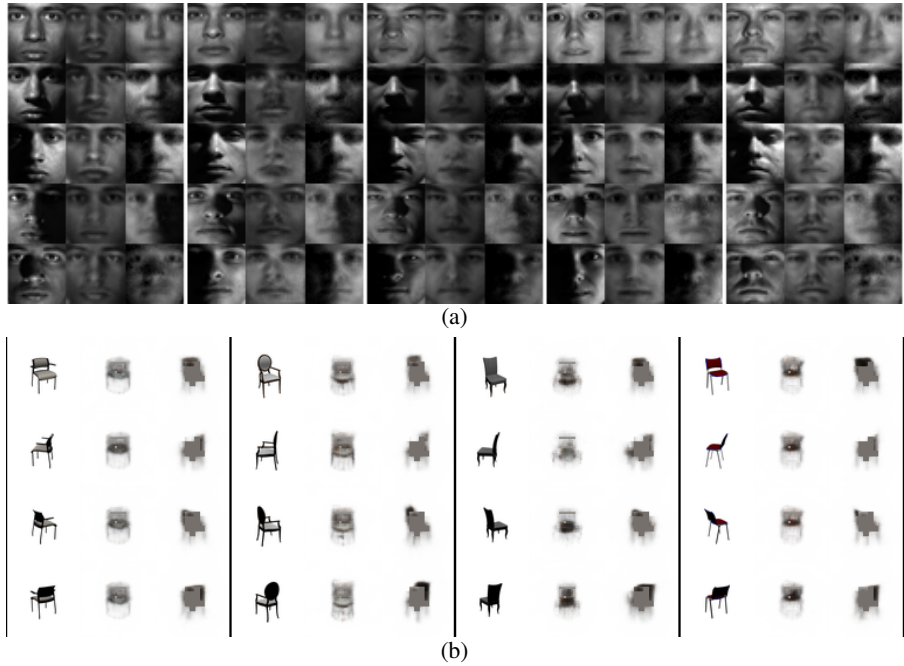

Figure 3: Reconstruction from $e_1$ and $e_2$ for (a) Extended Yale B and (b) Chairs. Columns in each block reflect (left to right): real, reconstruction from $e_1$ and that from $e_2$.

and lower-right, with the same split as $38 \times 5 = 190$ samples used for training and the rest used for testing [13, 14, 19]. We use the same architecture for the predictor and the encoder as CAI (as presented in [19]), i.e., single-layer neural networks, except that our encoder produces two encodings instead of one. We also model the decoder and the disentanglers as single-layer neural networks.

Table 1 summarizes the results. The proposed unsupervised method outperforms existing state-of-the-art (supervised) invariance induction methods on both $A_y$ and $A_z$ metrics, providing a significant boost on $A_y$ and complete removal of lighting information from $e_1$ reflected by $A_z$. Furthermore, the accuracy of predicting $z$ from $e_2$ is 0.89, which validates its automatic migration to $e_2$. Figure 2 shows t-SNE [15] visualization of raw data and embeddings $e_1$ and $e_2$ for our model. While raw data is clustered by lighting conditions $z$, $e_1$ exhibits clustering by $y$ with no grouping based on $z$, and $e_2$ exhibits near-perfect clustering by $z$. Figure 3a shows reconstructions from $e_1$ and $e_2$. Dedicated decoder networks were trained (with weights of $Enc$ frozen) to generate these visualizations. As evident, $e_1$ captures identity-related information but not lighting while $e_2$ captures the inverse.

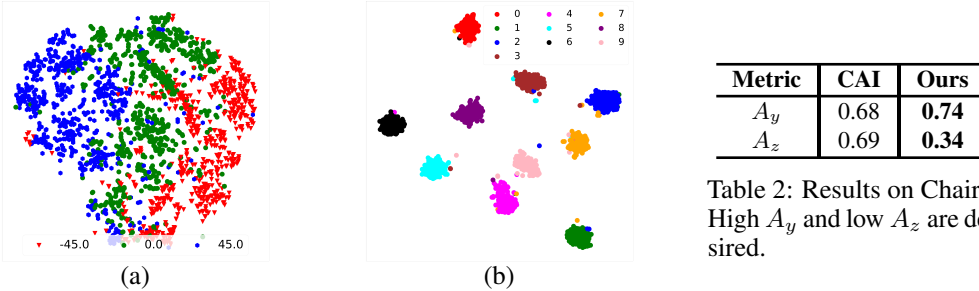

| Metric | CAI | Ours |
|--------|-----|------|
| $A_y$ | 0.68 | **0.74** |
| $A_z$ | 0.69 | **0.34** |

Table 2: Results on Chairs. High $A_y$ and low $A_z$ are desired.

(a)        (b)

Figure 4: MNIST-ROT – t-SNE visualization of (a) raw data and (b) $e_1$

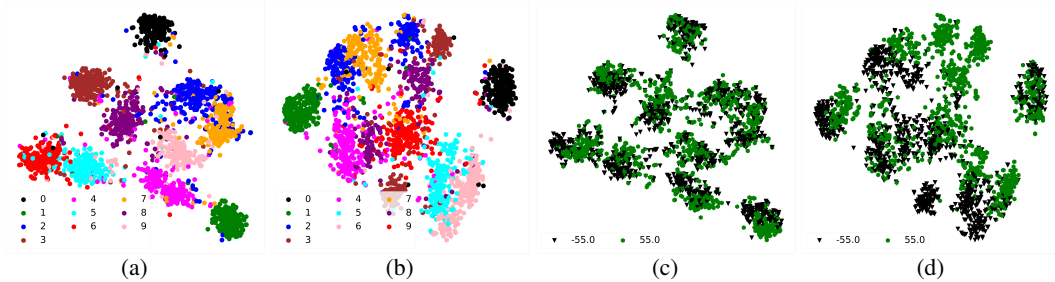

(a)     (b)     (c)     (d)

Figure 5: t-SNE visualization of MNIST-ROT $e_1$ embedding for the proposed Unsupervised Adversarial Invariance model (a) & (c), and baseline model $B_0$ (b) & (d). Models trained on $\Theta = \{0, \pm 22.5, \pm 45\}$. Visualization generated for $\Theta = \{\pm 55\}$.

**Chairs.** This dataset consists of 1393 different chair types rendered at 31 yaw angles and two pitch angles using a computer aided design model. We treat the chair identity as the target $y$ and the yaw angle $\theta$ as $z$. We split the data into training and testing sets by picking alternate yaw angles. Therefore, *there is no overlap of $\theta$ between the two sets*. We compare the performance of our model to CAI. In order to train the CAI model, we group $\theta$ into four categories – front, left, right and back, and provide it this information as a one-hot encoded vector. We model the encoder and the predictor as two-layer neural networks for both CAI and our model. We also model the decoder as a two-layer network and the disentanglers as single-layer networks. Table 2 summarizes the results, showing that our model outperforms CAI on both $A_y$ and $A_z$. Moreover, the accuracy of predicting $\theta$ from $e_2$ is 0.73, which shows that this information migrates to $e_2$. Figure 3b shows results of reconstructing $x$ from $e_1$ and $e_2$ generated in the same way as for Extended Yale-B above. The figure shows that $e_1$ contains identity information but nothing about $\theta$ while $e_2$ contains $\theta$ with limited identity information.

## 5.2 Effective use of synthetic data augmentation for learning invariance

Data is often not available for all possible variations of nuisance factors. A popular approach to learn models robust to such expected yet unobserved or infrequently seen (during training) variations is data augmentation through synthetic generation using methods ranging from simple operations [10] like rotation and translation to Generative Adversarial Networks [1, 8] for synthesis of more realistic

| Metric | Angle | CAI | Ours | $B_0$ | $B_1$ |
|---|---|---|---|---|---|
| $A_y$ | $\Theta$ | 0.958 | **0.977** | 0.974 | 0.972 |
| | $\pm 55^\circ$ | 0.826 | **0.856** | 0.826 | 0.829 |
| | $\pm 65^\circ$ | 0.662 | **0.696** | 0.674 | 0.682 |
| $A_z$ | - | 0.384 | **0.338** | 0.586 | 0.409 |

Table 3: Results on MNIST-ROT. $\Theta = \{0, \pm 22.5^\circ, \pm 45^\circ\}$ was used for training. High $A_y$ and low $A_z$ are desired.

| $k$ | CAI | Ours | $B_0$ | $B_1$ |
|---|---|---|---|---|
| -2 | 0.816 | **0.880** | 0.872 | 0.870 |
| 2 | 0.933 | **0.958** | 0.942 | 0.940 |
| 3 | 0.795 | **0.874** | 0.847 | 0.853 |
| 4 | 0.519 | **0.606** | 0.534 | 0.550 |

Table 4: MNIST-DIL – Accuracy of predicting $y$ ($A_y$). $k = -2$ represents erosion with kernel-size of 2.

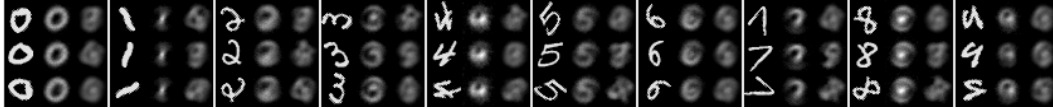

Figure 6: MNIST-ROT – reconstruction from $e_1$ and $e_2$, (c) $e$. Columns in each block reflect (left to right): real, reconstruction from $e_1$ and that from $e_2$.

variations. The prediction model is then trained on the expanded dataset. The resulting model, thus, becomes robust to specific forms of variations of certain nuisance factors that it has seen during training. Invariance induction, on the other hand, aims to completely prevent prediction models from using information about nuisance factors. Data augmentation methods can be more effectively used for improving the prediction of $y$ by using the expanded dataset for inducing invariance by *exclusion* rather than *inclusion*. We use two variants of the MNIST [12] dataset of handwritten digits to (1) show the advantage of unsupervised invariance induction at this task over its supervised variant through comparison with CAI, and (2) perform ablation experiments for our model to justify our framework design. We use the same two-layer architectures for the encoder and the predictor in both our model and CAI, except that our encoder generates two encodings instead of one. We model the decoder as a three-layer neural network and the disentanglers as single-layer neural networks. We train two baseline versions of our model for our ablation experiments – $B_0$ composed of $Enc$ and $Pred$, i.e., a single feed-forward network $x \to h \to y$ and $B_1$, which is the same as the composite model $M_1$, i.e., the proposed model trained non-adversarially without the disentanglers. $B_0$ is used to validate the phenomenon that invariance by exclusion is a better approach than robustness through inclusion whereas $B_1$ helps evaluate the importance of disentanglement in our framework.

**MNIST-ROT.** We create this variant of the MNIST dataset by randomly rotating each image by an angle $\theta \in \{-45^\circ, -22.5^\circ, 0^\circ, 22.5^\circ, 45^\circ\}$ about the Y-axis. We denote this set of angles as $\Theta$. The angle information is used as a one-hot encoding while training the CAI model. We evaluate all the models on the same metrics $A_y$ and $A_z$ we previously used. We additionally test all the models on $\theta \notin \Theta$ to gauge the performance of these models on unseen variations of the rotation nuisance factor. Table 3 summarizes the results, showing that our unsupervised adversarial model not only performs better than the baseline ablation versions but also outperforms CAI, which uses supervised information about the rotation angle. The difference in $A_y$ is especially notable for the cases where $\theta \notin \Theta$. Results on $A_z$ show that our model discards more information about $\theta$ than CAI even though CAI uses $\theta$ information during training. The information about $\theta$ migrates to $e_2$, indicated by the accuracy of predicting it from $e_2$ being $0.77$. Figure 4 shows t-SNE visualization of raw MNIST-ROT images and $e_1$ learned by our model. While raw data tends to cluster by the rotation angle, $e_1$ shows near-perfect grouping based on the digit-class. We further visualize the $e_1$ embedding learned by the proposed model and the baseline $B_0$, which models the classifier $x \to h \to y$, to investigate the effectiveness of invariance induction by exclusion versus inclusion, respectively. Both the models were trained on digits rotated by $\theta \in \Theta$ and t-SNE visualizations were generated for $\theta \in \{\pm 55\}$. Figure 5 shows the results. As evident, $e_1$ learned by the proposed model shows no clustering by the rotation angle, while that learned by $B_0$ does, with encodings of some digit classes forming multiple clusters corresponding to rotation angles. Figure 6 shows results of reconstructing $x$ from $e_1$ and $e_2$ generated in the same way as Extended Yale-B above. The figures show that reconstructions from $e_1$ reflect the digit class but contain no information about $\theta$, while those from $e_2$ exhibit the inverse.

**MNIST-DIL.** We create this variant of MNIST by eroding or dilating MNIST digits using various kernel-sizes ($k$). We use models trained on MNIST-ROT to report evaluation results on this dataset, to show the advantage of unsupervised invariance induction in cases where certain $z$ are not annotated

| Source - Target | DANN [6] | VFAE [14] | Ours |
|---|---|---|---|
| books - dvd | 0.784 | 0.799 | **0.820** |
| books - electronics | 0.733 | **0.792** | 0.764 |
| books - kitchen | 0.779 | **0.816** | 0.791 |
| dvd - books | 0.723 | 0.755 | **0.798** |
| dvd - electronics | 0.754 | 0.786 | **0.790** |
| dvd - kitchen | 0.783 | 0.822 | **0.826** |
| electronics - books | 0.713 | 0.727 | **0.734** |
| electronics - dvd | 0.738 | **0.765** | 0.740 |
| electronics - kitchen | 0.854 | 0.850 | **0.890** |
| kitchen - books | 0.709 | 0.720 | **0.724** |
| kitchen - dvd | 0.740 | 0.733 | **0.745** |
| kitchen - electronics | 0.843 | 0.838 | **0.859** |

Table 5: Results on Amazon Reviews dataset – Accuracy of predicting $y$ from $e_1$ ($A_y$)

in the training data. Thus, information about these $z$ cannot be used to train supervised invariance induction models. We also provide ablation results on this dataset using the same baselines $B_0$ and $B_1$. Table 4 summarizes the results of this experiment. The results show significantly better performance of our model compared to CAI and the baselines. More notably, CAI performs *significantly* worse than our baseline models, indicating that the supervised approach of invariance induction can worsen performance with respect to nuisance factors not accounted for during training.

## 5.3 Domain Adaptation

Domain adaptation has been treated as an invariance induction task in recent literature [6, 14] where the goal is to make the prediction task invariant to the "domain" information. We evaluate the performance of our model at domain adaptation on the Amazon Reviews dataset [4] using the same preprocessing as [14]. The dataset contains text reviews on products in four domains – "books", "dvd", "electronics", and "kitchen". Each review is represented as a feature vector of unigram and bigram counts. The target $y$ is the sentiment of the review – either positive or negative. We use the same experimental setup as [6, 14] where the model is trained on one domain and tested on another, thus creating 12 source-target combinations. We design the architectures of the encoder and the decoder in our model to be similar to those of VFAE, as presented in [14]. Table 5 shows the results of the proposed unsupervised adversarial model and supervised state-of-the-art methods VFAE and Domain Adversarial Neural Network (DANN) [6]. The results of the prior works are quoted directly from [14]. The results show that our model outperforms both VFAE and DANN at nine out of the twelve tasks. Thus, our model can also be used effectively for domain adaptation.

## 6 Conclusion And Future Work

In this paper, we have presented a novel unsupervised framework for invariance induction in neural networks. Our method models invariance as an information separation task achieved by competitive training between a predictor and a decoder coupled with disentanglement. We described an adversarial instantiation of this framework and provided analysis of its working. Experimental evaluation shows that our unsupervised adversarial invariance induction model outperforms state-of-the-art methods, which are supervised, on learning invariance to inherent nuisance factors, effectively using synthetic data augmentation for learning invariance, and domain adaptation. Furthermore, the fact that our framework requires no annotations for variations of nuisance factors, or even knowledge of such factors, shows the conceptual superiority of our approach compared to previous methods. Since our model does not make any assumptions about the data, it can be applied to any supervised learning task, eg., binary/multi-class classification or regression, without loss of generality.

The proposed approach is not designed to learn "fair representations" of data, e.g., making predictions about the savings of a person invariant to age, when such bias exists in data and making the prediction task invariant to such biasing factors is of higher priority than the prediction performance [19]. In future work, we will augment our model with the capability to *additionally* use supervised information (when available) about known nuisance factors for learning invariance to them, which will, consequently, help our model learn fair representations.

## Acknowledgements

This work is based on research sponsored by the Defense Advanced Research Projects Agency under agreement number FA8750-16-2-0204. The U.S. Government is authorized to reproduce and distribute reprints for governmental purposes notwithstanding any copyright notation thereon. The views and conclusions contained herein are those of the authors and should not be interpreted as necessarily representing the official policies or endorsements, either expressed or implied, of the Defense Advanced Research Projects Agency or the U.S. Government.

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
