[Reviews · NeurIPS 2018]

Reviewer 1



The paper proposes a method that explicitly separates relevant factors and irrelevant factors in a supervised learning task. The separation is unsupervised and combines several techniques to make it successful. It uses a reconstruction process to capture all information from the input. However, the latent representation is split into relevant ones and irrelevant (nuisance) factors. Only the relevant ones are used for prediction. For reconstruction, the relevant features are corrupted by noise in order to make them less reliable. An important step seems to be to make both parts of the latent space independent with an adversarial training. The paper is well written and the presentation is clear. Strong aspects: - important topic - relatively simple architecture - yields significant improvements - interesting adversarial training setup for independence - no additional information needed Weak aspects: - No standard deviations/statistics for the results given. How many independent runs did you do etc? - the comparison is mostly against CAI (probably the authors own prior work), except for the Yale-B and the Amazon Review task - the datasets/problems are relatively small/simple: Yale-B, Chairs, MNIST (with variants), Amazon Reviews. How does it perform on classification in the wild type, e.g. Imagenet or so? Do you expect any improvement or does it only work for fine-grained classification? - The weighting of the Loss terms introduces 2 hyperparameters, beta and gamma (however they were fixed for all experiments and rather generic). - it is not mentioned that the code for the paper will be published Line 211: Introduce z explicitly. E.g. as the "labeled nuisance factors" To summarize: the paper is interesting and makes a contribution. I would have liked to see technically more serious reporting with statistics, which I expect to happen in the final version. Also, a large-scale problem would have been nice.

Reviewer 2



The paper presents a general procedure to disentangle the factors of variation in the data that contribute to the prediction of a desired target variable and nuisance factors of variation. The authors achieve this by partitioning the hidden representations learned by neural network “h” into “e1” that contains factors that contribute directly to predicting the outcomes “y” and “e2” that contains nuisance factors that are not useful in predicting “y”. An adversarial disentanglement objective is used to ensure that e1 and e2 capture different pieces of information and a data reconstruction objective (with e1 noised) is used to ensure that all of the information is not present solely in e1 while e2 learns nothing. The overall approach is clever, sound and well thought out. The motivation of disentanglement and learning nuisance factors in data is well presented as well. My main concerns with this paper is the experimental work: 1) The paper claims to outperform methods where the nuisance factors are known in advance and can be used learn representations that are “explicitly” invariant to these factors via data augmentation or adversarial training. However, I think the baselines presented that use this information are not strong enough, for example in MNIST where different rotations or other affine transformations are applied to the data, how does this compare to the spatial transformer network? (Jaderberg et. al 2016). 2) I found almost all presented tasks to be toyish in nature, while it is certainly useful to experiment in settings where the nuisance factors are known in advance, I would encourage the authors to also work with richer datasets where an adversarial disentanglement objective, a classification objective and a reconstruction objective, working together can be hard to train. Specifically, I see the reconstruction objective becoming harder to optimize with high dimensional inputs. Demonstrating the capability of this model to work with richer datasets will help with adoption. Simple MNIST and YALE-B experiments are a good start but are insufficient to demonstrate clear improvements. 3) It would be informative to also report in Table 1 the accuracy of predicting y from e2 (by “post-fitting” a classifier after training) and z from e2. Along the same lines, more ablations that add different amounts of noise to e1 during reconstruction, alter the coefficients on the disentanglement versus reconstruction objectives would be useful to report. In conclusion I think that this is a promising piece of work that currently lacks strong experimental backing. I'd be more convinced by experiments on harder and richer datasets.

Reviewer 3



This paper presents an unsupervised invariance induction framework for neural networks that learns a split representation of data through competitive training between a prediction task and a reconstruction task. An adversarial instantiation of this framework and the corresponding analysis are provided. Experimental results on benchmark dataset are reported and discussed. Pros. 1. In general this paper is clearly organized and well written. The idea is novel and well motivated. An interesting unsupervised invariance induction framework is presented, with sufficient justifications. The technical details are easy to follow. 2. Empirical evaluations on different tasks are conducted and discussed. The proposed method achieves quite promising performance and outperforms many state-of-the-art algorithms. 3. Section 2 presents a very nice review of related work in the field. Cons. 1. One key component in the proposed framework is using a dropout layer as the noisy-transformer. More justifications should be provided. Other options for the noisy-transformer shall be discussed as well. 2. For experiments on domain adaptation reported in Section 5.3, two baselines DANN (JMLR, 2016) and VFAE (ICLR 2016) are employed. More recent baselines should be considered, such as [a]. [a] Zheng Li, Yu Zhang, Ying Wei, Yuxiang Wu, Qiang Yang: End-to-End Adversarial Memory Network for Cross-domain Sentiment Classification. IJCAI 2017: 2237-2243 ===== I have read the authors' response, and would like to keep my previous score.